# Expression of a single inhibitory member of the Ly49 receptor family is sufficient to license NK cells for effector functions

Sytse J Piersma[1,2]*, Shasha Li[1], Pamela Wong[1], Michael D Bern[1,3], Jennifer Poursine-Laurent[1], Liping Yang[1], Diana L Beckman[1], Bijal A Parikh[4], Wayne M Yokoyama[1,5]

[1]Division of Rheumatology, Department of Medicine, Washington University School of Medicine, St Louis, United States; [2]Siteman Cancer Center, Washington University School of Medicine, St Louis, United States; [3]Division of Oncology, Department of Medicine, Washington University School of Medicine, St Louis, United States; [4]Department of Pathology and Immunology, Washington University School of Medicine, St Louis, United States; [5]Bursky Center for Human Immunology and Immunotherapy Programs, Washington University School of Medicine, St. Louis, United States

*For correspondence:
spiersma@wustl.edu

## eLife Assessment

This study on mouse Ly49 receptors expressed on natural killer (NK) cells shows that Ly49A, in the presence of the corresponding MHC class I allele, can lead to NK cell licensing, thereby providing **valuable** insights into the mechanisms of NK cell modulation by Ly49 receptors. The work may have significant implications for studies of human killer-cell immunoglobulin-like receptors (KIR) expressing and other NK cells. Overall, the study was well-developed with **convincing** evidence.

**Abstract** Natural killer (NK) cells recognize target cells through germline-encoded activation and inhibitory receptors enabling effective immunity against viruses and cancer. The Ly49 receptor family in the mouse and killer immunoglobin-like receptor family in humans play a central role in NK cell immunity through recognition of major histocompatibility complex class I (MHC-I) and related molecules. Functionally, these receptor families are involved in the licensing and rejection of MHC-I-deficient cells through missing-self. The Ly49 family is highly polymorphic, making it challenging to detail the contributions of individual Ly49 receptors to NK cell function. Herein, we showed mice lacking expression of all Ly49s were unable to reject missing-self target cells in vivo, were defective in NK cell licensing, and displayed lower KLRG1 on the surface of NK cells. Expression of Ly49A alone on an H-2D$^d$ background restored missing-self target cell rejection, NK cell licensing, and NK cell KLRG1 expression. Thus, a single inhibitory Ly49 receptor is sufficient to license NK cells and mediate missing-self in vivo.

## Introduction

Natural killer (NK) cells are innate lymphoid cells (ILCs) that can mediate effective immunity against viruses and cancer through direct lysis and cytokine production (*Huntington et al., 2020*; *Piersma and Brizić, 2022*). NK cells recognize their target cells through integration of signals by germline-encoded activation and inhibitory receptors (*Long et al., 2013*). These inhibitory receptors include members of

**eLife digest** Immune cells help to protect the body from infections and cancer. One way that they do this is by recognising and destroying harmful cells. Natural killer cells (known as NK cells) can tell whether cells are healthy or harmful based on the molecules that are present on their surface. For example, molecules known as MHC Class I are commonly found on healthy cells whereas cancer cells and those infected with viruses can lose these molecules from their surface.

NK cells use inhibitory receptors to recognise MHC Class I molecules. If they do not detect this molecule on the surface of a cell, the NK cell can become activated and kill the unhealthy cell. In mice, a group of receptors known as the Ly49 family are thought to be responsible for this process. However, Ly49 receptors are complex and their individual roles in NK cells remained unclear.

To investigate, Piersma et al. developed mice without any of the Ly49 receptors. A single receptor was then re-introduced to identify its individual role. These experiments showed that the Ly49 receptor family is required for NK cells to detect a lack of MHC Class I molecules on cells. The receptors also allow NK cells to express certain receptors and to be fully functional. Introducing an individual Ly49 receptor was sufficient to restore these abilities, suggesting that a single receptor can mediate these processes.

The findings of Piersma et al. provide a deeper understanding of how NK cell inhibitory receptors work in mice. Further investigation of the equivalent receptors in humans may aid understanding of the mechanisms involved in recognising harmful changes to cells. In the future, this information may help to improve cancer and anti-viral treatments.

the Ly49 family in the mouse that are encoded by the *Klra* gene family and killer immunoglobin-like receptor (KIR) family in humans and they prevent killing of healthy cells through recognition of major histocompatibility complex class I (MHC-I) (*Colonna and Samaridis, 1995*; *Karlhofer et al., 1992*). Host cells may lose surface MHC-I expression in response to virus infection or malignant transformation. As a result, these cells become invisible to CD8$^+$ T cells, but simultaneously become targets for NK cells through 'missing-self' recognition (*Kärre et al., 1986*).

The inhibitory Ly49 molecules appear to be responsible for missing-self recognition in mice (*Babić et al., 2010*; *Bélanger et al., 2012*; *Gamache et al., 2019*; *Parikh et al., 2020*; *Zhang et al., 2019*). However, it has been sometimes challenging to draw definitive conclusions because the Ly49 family is highly polymorphic and differs between mouse strains. Moreover, multiple inhibitory Ly49 receptors within a single host can recognize a given MHC-I molecule while others apparently have no ligands and instead recognize other MHC-I alleles (*Schenkel et al., 2013*). Yet, the Ly49s for non-host MHC-I alleles are still expressed. For example, in the C57BL/6 background, Ly49C and Ly49I can recognize H-2$^b$ MHC-I molecules that include H-2K$^b$ and H-2D$^b$, while Ly49A and Ly49G cannot recognize H-2$^b$ molecules and instead they recognize H-2$^d$ alleles. Still these Ly49s are expressed in C57BL/6 mice, so their individual contributions to missing-self rejection are unclear. Ly49A has also been implicated in recognition of the non-classical MHC-I molecule H2-M3, which is upregulated in response to exposure to N-formylated peptides (*Andrews et al., 2012*; *Chiu et al., 1999*). Importantly, the specificities of several Ly49s have been clearly established while others remain to be confirmed. For example, the binding of Ly49A to H-2D$^d$ has been confirmed by crystallographic studies (*Tormo et al., 1999*), and validated by mutational analysis of both Ly49A and H-2D$^d$ (*Matsumoto et al., 2001*; *Wang et al., 2001*). By contrast, the MHC-I specificities of other Ly49s have been primarily studied with MHC tetramers containing human β$_2$-microglobulin (B2m), which is not recognized by Ly49A (*Mitsuki et al., 2004*), on cells overexpressing Ly49s (*Hanke et al., 1999*). Thus, the contributions of individual Ly49 receptors to NK cell effector function are confounded by the expression of multiple receptors, some of which may be irrelevant to a given self-MHC haplotype, and multiple Ly49 molecules whose specificities are less well defined.

In addition to effector function in missing-self, Ly49 receptors that recognize their cognate MHC-I ligands are involved in the licensing or education of NK cells to acquire functional competence. NK cell licensing is characterized by potent effector functions including IFNγ production and degranulation in response to activation receptor stimulation (*Elliott et al., 2010*; *Kim et al., 2005*). Like missing-self recognition, inhibitory Ly49s require SHP-1 for NK cell licensing, which interacts with the

ITIM-motif encoded in the cytosolic tail of inhibitory Ly49s (*Bern et al., 2017*; *Kim et al., 2005*; *Viant et al., 2014*). Moreover, lower expression of SHP-1, particularly within the immunological synapse, is associated with licensed NK cells (*Schmied et al., 2023*; *Wu et al., 2021*). Thus, inhibitory Ly49s have a second function that licenses NK cells to self-MHC-I, thereby generating functionally competent NK cells, but it has not been possible to exclude contributions from other co-expressed Ly49s.

The complex nature of the Ly49 family confounds our fundamental understanding of these receptors, particularly regarding their function in vivo. To better understand Ly49 function in vivo, several groups made mutant mouse lines with altered expression of the *Klra* locus. This was pioneered by the Makrigiannis group, which targeted the promotor of *Klra15* encoding Ly49O in 129 ES cells and resulting mice were subsequently backcrossed to C57BL/6 background (*Bélanger et al., 2012*). These mice were defective in the rejection of MHC-I-deficient target cells in vivo and exhibited reduced tumor control as well (*Tu et al., 2014*). However, there was limited surface expression of NKG2A as well as Ly49s so these target defects were not solely dependent on the absence of Ly49s. Moreover, these mice likely carried 129 alleles of other genes in the NK gene complex (NKC) that are expressed on NK cells, display allelic polymorphisms and are genetically linked to *Klra* locus that encode the Ly49s. Following the development of mouse CRISPR engineering, the Dong group deleted the entire 1.4 Mb *Klra* locus in the C57BL/6 NKC and showed that Ly49-deficient mice were unable to reject MHC-I-deficient cells in vivo at steady state (*Zhang et al., 2019*). Besides loss of Ly49 receptor expression, surface expression of other receptors, including KLRG1, NKG2A, NKG2D, and CD94, was also reduced in these mice. Around the same time, we generated a mouse that contains a 66 kb and 149 kb deletion in the C57BL/6 *Klra* locus, resulting in the loss of four Ly49 molecules, including Ly49A and Ly49G (*Parikh et al., 2020*). The resulting ΔLy49-1 mice were also deficient in H2D$^d$-restricted control of murine cytomegalovirus (MCMV), which was rescued by knock-in of Ly49A into the *Ncr1* locus. Thus, available genetic evidence suggests inhibitory Ly49 receptors are essential for licensing and missing-self recognition.

The current data, however, do not take into account that individual Ly49s are stochastically expressed on NK cells, and multiple receptors are simultaneously expressed on individual NK cells, resulting in a diverse Ly49 repertoire of potential specificities on overlapping subsets of NK cells (*Dorfman and Raulet, 1998*; *Kubota et al., 1999*; *Smith et al., 2000*). As a result, not all NK cells express a specific Ly49 receptor and most NK cells express multiple Ly49 molecules, making it difficult to study the biology of a specific Ly49 receptor without genetic approaches. However, *Klra* genes encoding the Ly49 receptor family are highly related and clustered together, resulting in a high concentration of repetitive elements (*Makrigiannis et al., 2005*), complicating the capacity to target individual *Klra* genes for definitive analysis. Moreover, in ΔLy49-1 mice, which had an intact *Klra4* coding sequence encoding Ly49D, the percentage of NK cells expressing Ly49D was markedly reduced, even though Ly49D was otherwise expressed at normal levels, suggesting a regulatory locus control region within the deleted fragments (*Parikh et al., 2020*). Such data raise the possibility that the large genetic deletions of the Ly49 locus may affect other NK cell receptors in the NKC that contribute to NK cell function. Thus, expression of individual Ly49 receptors without confounding effects of other Ly49s and potentially other NKC genes is needed to validate the conclusions from the study of mice lacking Ly49 expression.

Here, we studied the role of an individual Ly49 receptor in NK cell function. To this end, we deleted all NK cell *Klra* genes encoding the Ly49 receptor family using CRISPR/Cas9 and confirmed the role of the Ly49 family in missing self and licensing. Subsequently, we expressed Ly49A in isolation under control of the *Ncr1* locus in Ly49-deficient mice on an H-2D$^d$ background to show that a single inhibitory Ly49 receptor expressed by all NK cells is sufficient for licensing and mediating missing-self in vivo.

## Results

### NK cells from CRISPR-generated mice lacking all NK cell-related Ly49 molecules display reduced KLRG1 expression

To investigate the role of individual Ly49 molecules, we generated a mouse that lacked all expressed Ly49 receptors. We targeted the remaining *Klra* region encoding the Ly49 receptor family in our previously published ΔLy49-1 mouse (*Parikh et al., 2020*) with guide RNAs targeting *Klra9* and *Klra17*.

The resulting mouse contained a fusion between *Klra9* and *Klra17* with a deletion of the start codon and insertion of a fusion sequence that did contain a potential start site for a putative five amino acid polypeptide (*Figure 1A*). We confirmed that the 3′ deletion reported in the ΔLy49-1 mouse was unaffected, resulting in a frameshift and a premature stop codon after nine amino acids in the fused *Klra7/1* gene. Thus, genetic and sequencing analysis revealed all Ly49 genes were disrupted and we termed this mouse line Ly49KO (*Figure 1A*).

Flow cytometry confirmed loss of cell surface expression of Ly49 molecules in homozygous Ly49KO mice (*Figure 1B*). NKG2A, CD94, and NKG2D molecules that are encoded by the NKG2 locus, located next to the Ly49 locus, were still expressed, albeit at marginally lower frequencies (*Figure 1C*; *Yokoyama and Plougastel, 2003*). Unrelated molecules 2B4 and CD122 were unaffected. In heterozygous Ly49KO mice, the percentages of NK cells expressing Ly49A, Ly49C, Ly49D, Ly49G2, Ly49H, and Ly49I were reduced by 33–41%. The median fluorescent intensity (MFI) for Ly49I was reduced by 26% in Ly49I+ NK cells in heterozygous Ly49KO mice, while the other Ly49s did not display significant differences in MFI. Consistent with apparent dependence of normal KLRG1 expression on MHC-I expression (*Corral et al., 2000*) and previous reports (*Zhang et al., 2019*), Ly49KO NK cells showed a 51% reduction in KLRG1 expression (*Figure 1C*). NK cells in Ly49KO mice displayed similar maturation to wildtype NK cells, based on the expression of the surface markers CD27 and CD11b (*Figure 1D*). Thus, Ly49KO mice specifically lack all Ly49 molecules and display moderate alterations in select surface molecules while showing otherwise normal numbers of apparently mature NK cells.

## Ly49-deficient NK cells are defective in licensing and rejection of MHC-I-deficient target cells

Inhibitory Ly49-positive NK cells can be licensed through recognition of cognate MHC-I molecules, resulting in a phenotype of increased IFNγ production following plate-bound anti-NK1.1 stimulation (*Kim et al., 2005*). NKG2A has been implicated in NK cell licensing by the non-classical MHC-I molecule Qa1 (*Anfossi et al., 2006*), and to eliminate potential confounding effects by this interaction, effector functions of NKG2A- NK cells were evaluated as described before (*Bern et al., 2017*). Stimulation of Ly49KO NK cells with anti-NK1.1 resulted in a 73% reduction in IFNγ production compared to wildtype NK cells, similar to unlicensed NK cells from H-2K$^b$ and H-2D$^b$ double-deficient (KODO) mice (*Figure 2A*). Both Ly49KO and KODO NK cells produced high amounts of IFNγ in response to phorbol 12-myristate 13-acetate (PMA) plus ionomycin (*Figure 2B*), indicating that their IFNγ production machinery is intact. Thus, these results confirm that Ly49 molecules are required for the NK cell licensed phenotype.

To investigate the capability of Ly49KO NK cells to reject MHC-I-deficient target cells, we challenged anti-NK1.1 NK cell-depleted, wildtype, and Ly49KO mice with a mixture of wildtype, B2m-deficient × KODO (MHC-I KO), H-2D$^b$ KO, and H-2K$^b$ KO splenocytes that were differentially labeled with CellTrace Far Red (CTFR) and CellTrace Violet (CTV) (*Figure 2C*). While wildtype mice efficiently rejected MHC-I KO and H-2K$^b$ KO target cells, only 13% of H-2D$^b$ KO target cells were rejected by wildtype mice, indicating that missing-self recognition in the C57BL/6 background depends on the absence of H-2K$^b$ rather than H-2D$^b$. None of the target cell populations were rejected in the Ly49KO mice, comparable to wildtype controls depleted of NK cells. Thus, Ly49 molecules mediate NK cell-dependent MHC-I-deficient target cell killing in vivo under steady-state conditions and rejection of MHC-I-deficient target cells is predominantly controlled by H-2K$^b$ in the H-2$^b$ background.

## Expression of Ly49A in Ly49-deficient H-2D$^d$ transgenic mice rescues KLRG1 expression

To investigate the potential of a single inhibitory Ly49 receptor on mediating NK cell licensing and missing-self rejection, the Ly49KO mice were backcrossed to H-2D$^d$ transgenic KODO (D8-KODO) Ly49A KI mice that express *Klra1* cDNA encoding the inhibitory Ly49A receptor in the *Ncr1* locus encoding NKp46 and its cognate ligand H-2D$^d$ but not any other classical MHC-I molecules (*Parikh et al., 2020*). Ly49A expression in the resulting Ly49KO/Ly49A KI D8-KODO mice closely follows NKp46 expression because NKp46- NK1.1+ NK cells in the bone marrow of these mice do not express Ly49A, while virtually all the NKp46+ NK1.1+ NK cells in the bone marrow and spleen express Ly49A (*Figure 3A*). Ly49KO/Ly49A KI D8-KODO NK cells expressed robust levels of Ly49A, albeit at lower MFI compared to Ly49A expression on D8-KODO NK cells, consistent with prior observations with

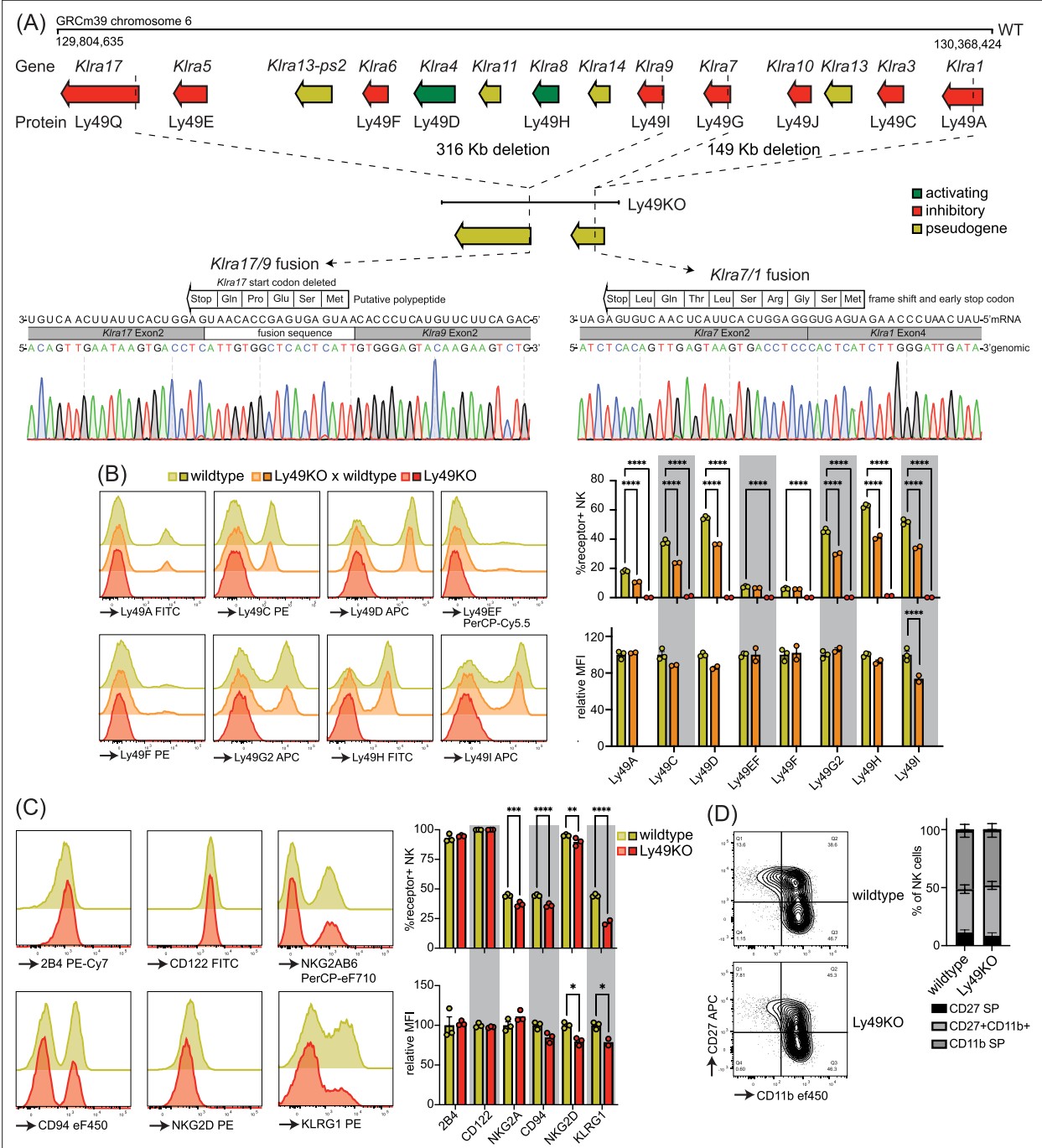

**Figure 1.** Mice generated to lack all natural killer (NK)-related Ly49 molecules using CRISPR have NK cells that display alterations in select surface molecules. (**A**) Genetic map of the Ly49 family encoded by the *Klra* gene locus of wildtype C57BL/6 and Ly49KO mice with Sanger sequencing of the fusion sequences in the Ly49KO mice. (**B**) Ly49 receptor expression on splenic NK cells of the indicated genotype with 2-3 mice per group. (**C**) Surface receptor expression on splenic NK cells from indicated genotype with 3 mice per group. (**D**) Expression of the maturation markers CD27 and CD11b on splenic NK cells from indicated genotype with 3 mice per group. MFI, median fluorescent intensity. Statistics were calculated using two-way ANOVA with corrections for multiple testing. Error bars indicate SEM; ns, not significant; *p<0.05, **p<0.01, ***p<0.001, and ****p<0.0001.

The online version of this article includes the following source data for figure 1:

**Source data 1.** Ly49KO Sanger sequencing files.

**Source data 2.** Receptor expression profile raw data.

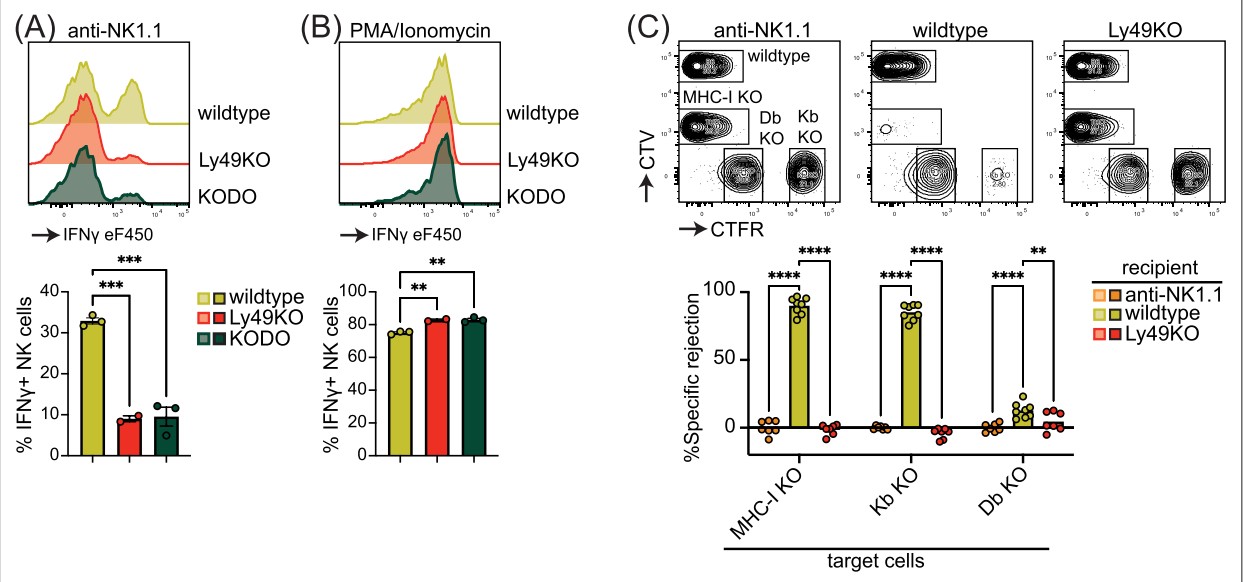

**Figure 2.** Natural killer (NK) cell licensing and rejection of MHC-I-deficient target cells are defective in Ly49KO mice. Splenocytes from the indicated mice were stimulated with plate-bound anti-NK1.1 (**A**) or phorbol 12-myristate 13-acetate (PMA)/ionomycin (**B**) and IFNγ production by NKG2A- NK cells and analyzed by flow cytometry with 2-3 mice per group. (**C**) In vivo cytotoxicity assay against H-2K$^b$, H-2D$^b$, and full MHC-I -deficient targets. Splenocytes from WT, H-2Kb, H-2Db, and MHC-I-deficient mice were differentially labeled with CellTrace Violet (CTV) and CellTrace Far Red (CTFR) as indicated. Mixture of labeled target cells was injected i.v. into wildtype, Ly49KO, and anti-NK1.1-depleted mice with 6-8 mice per group. Target cells were analyzed in the spleens by flow cytometry 2 days after challenge. KODO, H-2K$^b$ × H-2D$^b$ knock out; MHC-I KO, KODO × B2m knockout; MFI, median fluorescent intensity. Statistics were calculated using one-way (A and B) and two-way (C) ANOVA with corrections for multiple testing. Error bars indicate SEM; ns, not significant; *p<0.05, **p<0.01, ***p<0.001, and ****p<0.0001.

The online version of this article includes the following source data for figure 2:

**Source data 1.** IFNγ production and in vivo killing assay raw data.

wildtype Ly49A and H2D$^d$ (**Held et al., 1996**; **Karlhofer et al., 1994**). NK cells were able to fully mature in Ly49KO D8-KODO and Ly49KO/Ly49A KI D8-KODO mice as we observed similar percentage of mature CD27$^-$ CD11b$^+$ NK cells in the spleen, bone marrow, and liver (**Figure 3B**). While there was a modest significant increase in immature CD27$^+$ CD11b$^-$ NK cells in the bone marrow of Ly49KO D8-KODO and Ly49KO/Ly49A KI D8-KODO mice, no differences were observed in NK cell maturation in the spleen and liver. The decrease in the frequency of KLRG1$^+$ NK cells observed in the Ly49KO NK cells on the H-2$^b$ background was recapitulated in the D8-KODO background (**Figure 3C**). Intriguingly, expression of Ly49A in Ly49KO/Ly49A KI D8-KODO mice rescued KLRG1 expression and resulted in similar levels of KLRG1 as D8-KODO NK cells. Taken together, Ly49A engineered to be encoded within the NKp46 locus was efficiently expressed as an isolated Ly49 receptor and supported KLRG1 expression.

## NK cells expressing Ly49A in isolation are fully licensed and capable of rejecting MHC-I-deficient target cells

Next, we investigated the potential of Ly49A expression alone to mediate NK cell licensing and rejection of MHC-I-deficient target cells. In D8-KODO mice, Ly49A$^+$ NK cells displayed increased levels of IFNγ production and degranulation measured by CD107 in response to plate-bound anti-NK1.1 stimulation compared to all NK cells including unlicensed cells (**Figure 4A**). Similar to Ly49KO NK cells on the H-2$^b$ background, Ly49KO NK cells on the D8-KODO background showed a 72% decrease in IFNγ production, but also showed a 59% decrease in degranulation compared to Ly49A$^+$ NK cells in D8-KODO mice. This impaired IFNγ production and degranulation was reversed in Ly49A KI NK cells on the Ly49KO D8-KODO background that showed similar IFNγ production and degranulation to Ly49A$^+$ NK cells on the D8-KODO background, indicating that Ly49A KI NK cells are licensed. Importantly, there were no differences among all NK cell populations in IFNγ production and degranulation in response to PMA/Ionomycin (**Figure 4B**), indicating that the IFNγ production and degranulation

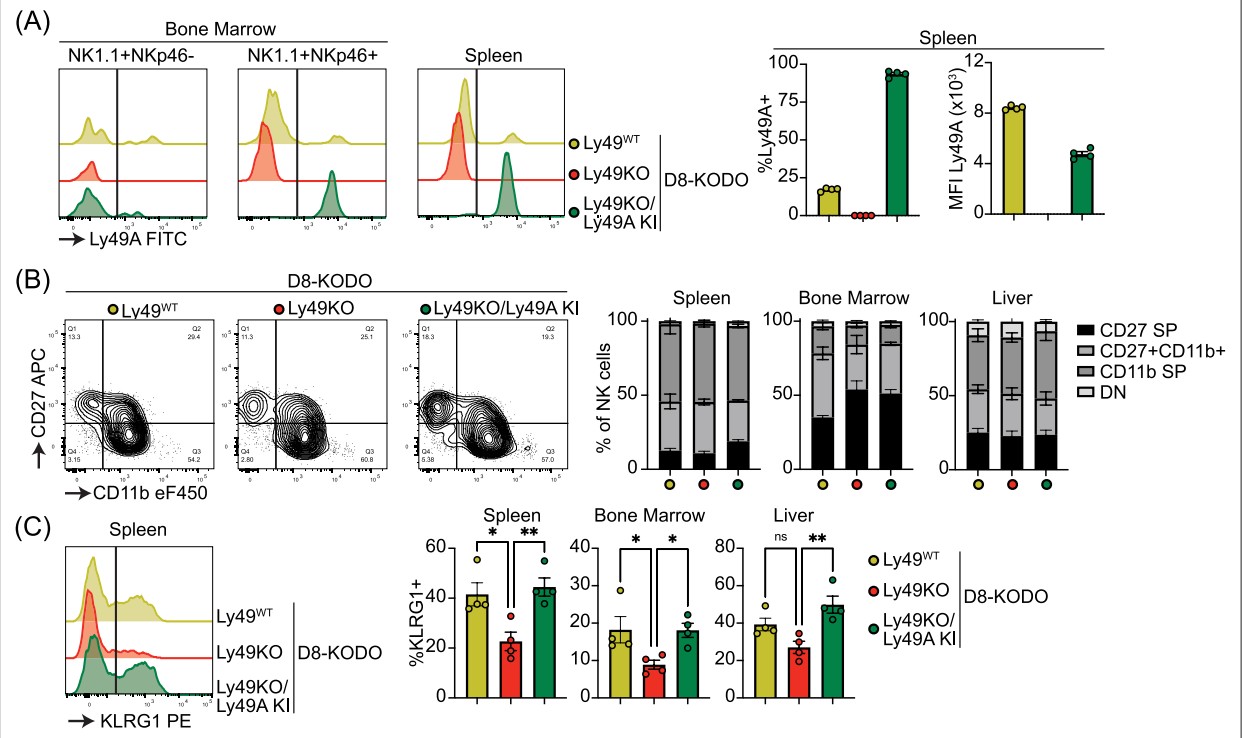

**Figure 3.** Ly49A is efficiently expressed in Ncr1-Ly49A knock-in mice and rescues KLRG1 expression in natural killer (NK) cells. Flow cytometric analysis of NK cells in D8-KODO, Ly49KO D8-KODO, and Ly49KO/Ly49A KI D8-KODO mice (**A**) Ly49A expression in NKp46[+] and NKp46[−] NK1.1[+] NK cells in bone marrow and spleen of the indicated mice. (**B**) Expression of the maturation markers CD27 and CD11b on NK cells in the spleen, bone marrow, and liver of indicated mice. (**C**) KLRG1 expression by NK cells in the spleen, bone marrow, and liver of indicated mice. MFI, median fluorescent intensity. All data is from 4 mice per group. Statistics were calculated using one-way ANOVA with corrections for multiple testing. Error bars indicate SEM; ns, not significant; *p<0.05, **p<0.01, ***p<0.001, and ****p<0.0001.

The online version of this article includes the following source data for figure 3:

**Source data 1.** Receptor expression profile raw data.

machinery are not affected in any of the mouse strains. Thus, the Ly49KO/Ly49A KI D8-KODO NK cells displayed a fully licensed phenotype comparable to licensed Ly49A[+] NK cells in D8-KODO mice.

Finally, we interrogated whether a single inhibitory Ly49 molecule would be sufficient to mediate missing-self rejection of MHC-I-deficient cells. To this end, D8-KODO, Ly49KO D8-KODO, and Ly49KO/Ly49A KI D8-KODO mice were challenged with a mixture of D8-KODO and KODO (MHC-I-deficient) splenocytes that were differentially labeled with CTV. D8-KODO mice efficiently rejected KODO target splenocytes, while Ly49KO D8-KODO mice were unable to reject these cells. However, this defect was completely restored in the Ly49KO/Ly49A KI D8-KODO mice, demonstrating that a single inhibitory Ly49 receptor is sufficient to mediate missing-self rejection of cells lacking its MHC class I ligand.

## Discussion

Several mice with genetic modifications in the Ly49 complex have been developed to study the role of Ly49 receptors but they have limitations (*Bélanger et al., 2012*; *Bern et al., 2017*; *Gamache et al., 2019*; *Parikh et al., 2020*; *Zhang et al., 2019*). A complicating factor in these studies is that multiple Ly49s, often with incompletely understood specificities, may be involved in target cell recognition. Here, we studied a mouse where a single Ly49 was under control of the *Ncr1* locus, which is expressed on all NK cells on the background of a complete Ly49 KO. Consistent with previous reports (*Bélanger et al., 2012*; *Parikh et al., 2020*; *Zhang et al., 2019*), Ly49-deficient NK cells were deficient in licensing and missing-self rejection, both on a H-2[b] background and in the presence of a single classical MHC-I allele, H-2D[d]. While our results did not interrogate licensing by inhibitory

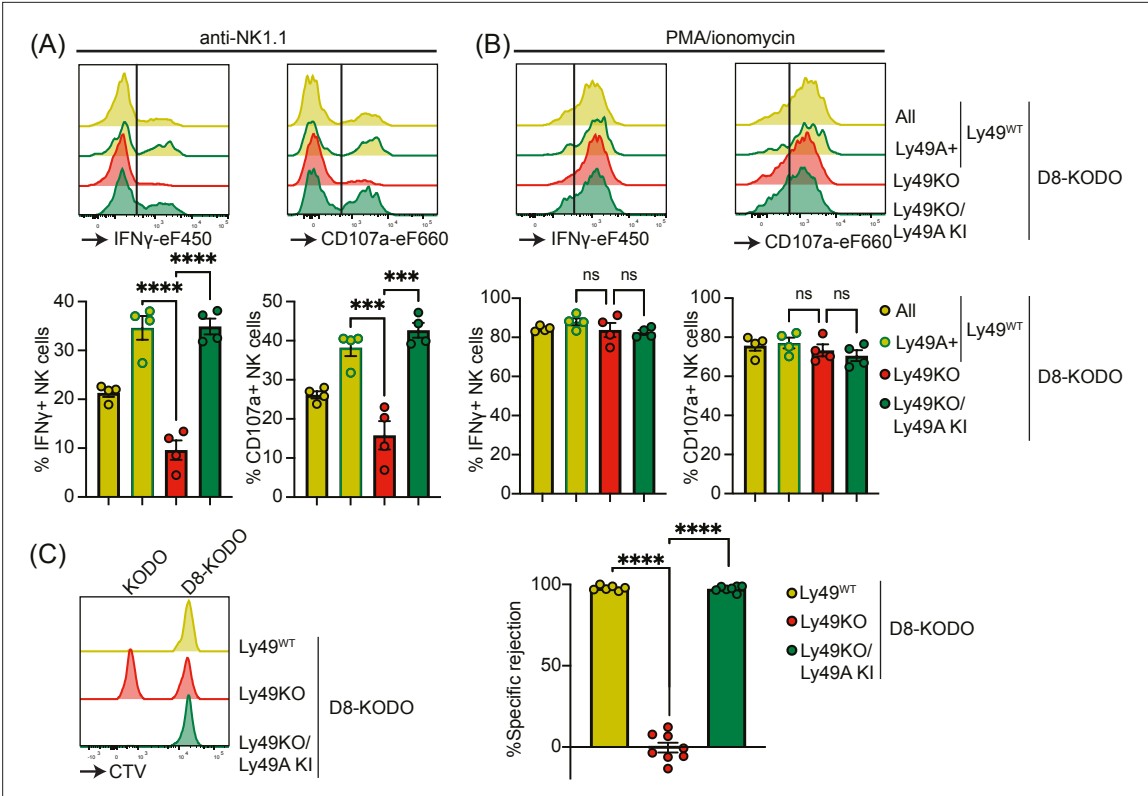

**Figure 4.** Expression of Ly49A in isolation is sufficient for natural killer (NK) cell licensing and missing-self rejection. Splenocytes from the indicated mice were stimulated with plate-bound anti-NK1.1 (**A**) or phorbol 12-myristate 13-acetate (PMA)/ionomycin (**B**) with 4 mice per group. IFNγ production and degranulation (CD107a) by NKG2A- NK cells were analyzed by flow cytometry. (**C**) Splenocytes from D8-KODO and KODO mice were differentially labeled with CTV as indicated. A mixture of labeled target cells was injected i.v. into D8-KODO, Ly49KO D8-KODO, and Ly49KO/Ly49A KI D8-KODO mice with 6-8 mice per group. Specific rejection of target cells was analyzed in spleens by flow cytometry 2 days after challenge. Statistics were calculated using one-way ANOVA with corrections for multiple testing. Error bars indicate SEM; ns, not significant; ***p<0.001 and ****p<0.0001.

The online version of this article includes the following source data for figure 4:

**Source data 1.** IFNγ production, degranulation, and in vivo killing assay raw data.

receptors outside of the Ly49 receptor family, such as has been reported for NKG2A (*Anfossi et al., 2006*; *Zhang et al., 2019*), they do demonstrate that expression of Ly49A without other Ly49 family members can mediate NK cell licensing. Moreover, we found that Ly49 receptors are required and sufficient for missing-self rejection under steady-state conditions. However, these observations do not rule out the involvement of other inhibitory receptors under specific inflammatory conditions. For example, NKG2A contributes to the rejection of missing-self targets in poly(I:C)-treated mice (*Zhang et al., 2019*). Finally, expression of the inhibitory Ly49A in isolation did not alter NK cell numbers or maturation, indicating that the Ly49s do not affect these parameters of NK cells. Yet the NK cells were fully licensed in terms of IFNγ production and degranulation in vitro and efficiently rejected MHC-I-deficient target cells in vivo. Thus, a single Ly49 receptor is capable of conferring the licensed phenotype and missing-self rejection in vitro and in vivo.

We observed that rejection of H-2K$^b$-deficient targets was more potent than H-2D$^b$-deficient target splenocytes, comparable to previous observations (*Johansson et al., 2005*). These data indicate that H-2K$^b$ is more efficiently recognized by Ly49s compared to H-2D$^b$. This is further supported by early studies using Ly49 transfectants binding to Con A blasts showing that Ly49C and Ly49I can bind to H-2D$^b$-deficient but not H-2K$^b$-deficient cells (*Hanke et al., 1999*), despite the caveat of testing binding to cells overexpressing Ly49s in these studies. Our studies indicate that the efficiency of Ly49-dependent missing-self rejection depends on the characteristics of specific MHC-I alleles recognized by cognate Ly49 receptors in vivo.

KLRG1 is an inhibitory receptor that recognizes E-, N-, and R- cadherins to inhibit NK cell cytotoxicity (*Ito et al., 2006*). KLRG1 is expressed on a subset of mature NK cells and can be upregulated in response to proliferation in a host with lymphopenia (*Huntington et al., 2007*). Consistent with previously published results (*Zhang et al., 2019*), we observed decreased KLRG1 expression in Ly49-deficient NK cells. The *Klra* gene family as well as *Klrg1* is located within the NKC on chromosome 6 (*Yokoyama and Plougastel, 2003*), thus an effect of regulatory elements deleted in Ly49KO mice cannot be excluded. This is further emphasized by studies of our ΔLy49-1 mouse, which expresses Ly49D on fewer NK cells (*Parikh et al., 2020*). The *Klra4* gene appears intact and Ly49D$^+$ NK cells expressed Ly49D at normal levels, suggesting the absence of a regulatory element in the deleted regions. Here, we showed the expression of only Ly49A, encoded in the *Ncr1* locus located on chromosome 7, in Ly49KO mice on an H-2D$^d$ background restored KLRG1 expression in NK cells from different tissues, indicating that inhibitory Ly49 receptors rather than regulatory elements influence KLRG1 expression. Moreover, NK cell KLRG1 expression is modulated by MHC-I molecules (*Corral et al., 2000*). Therefore, KLRG1 expression may be modulated as a consequence of NK cell licensing through inhibitory Ly49 receptors.

The Ly49 family is stochastically expressed in NK cells, resulting in an NK cell repertoire with different combinations of Ly49 receptors on individual NK cells. Not all Ly49 alleles are equally expressed, which has been suggested to be dependent on allelic exclusion (*Held et al., 1995*). Epigenetic control, including DNA-methylation, histone modification, and regulatory elements, has been linked to Ly49 expression (*Kissiov et al., 2022*; *McCullen et al., 2016*; *Rouhi et al., 2006*; *Saleh et al., 2004*). We observed that in Ly49KO heterozygous mice the percentage of Ly49$^+$ NK cells was reduced for each Ly49 molecule, yet the expression level measured by MFI was not affected except for Ly49I, indicating that loss of one allele does not affect Ly49 surface levels. Nonetheless, alternate mechanisms may control Ly49 expression as was observed for Ly49I. Expression of *Klra1* encoding Ly49A within the *Ncr1* locus resulted in ubiquitous Ly49A expression in NK cells, albeit at lower levels compared to Ly49A$^+$ D8-KODO NK cells. Despite the lower expression levels of Ly49A, Ly49A KI NK cells were fully licensed and efficiently eliminated MHC-I-deficient target cells, suggesting that minor alterations in expression levels of various Ly49s on individual NK cells may not affect NK cell functions. In conclusion, these data show that the expression of a single inhibitory Ly49 receptor is necessary and sufficient to license NK cells and mediate missing self-rejection under steady-state conditions in vivo.

## Materials and methods
### Animals
C57BL/6 (stock# 556) mice were purchased from Charles Rivers laboratories, B2m-deficient (stock# 2087) were purchased from Jackson Laboratories, H-2K$^b$ × H-2D$^b$ double-deficient (stock# 4215; KODO) mice were purchased from Taconic Farms. D8 is an H-2D$^d$ transgenic mouse that has been previously described (*Bieberich et al., 1986*) and was provided by D. Marguiles, National Institute of Allergy and Infectious Diseases, Bethesda, MD. D8-KODO mice have been previously generated by crossing D8 transgenic mice to a KODO background (*Choi et al., 2011*). ΔLy49-1 and Ly49A KI mice were previously generated in our laboratory (*Parikh et al., 2020*). In Ly49A KI mice, the stop codon of Ncr1 encoding NKp46 is replaced with a P2A peptide-cleavage site upstream of the Ly49A cDNA, while maintaining the 3′ untranslated region. All mice were maintained within the Washington University animal facility in accordance with the institutional ethical guidelines under protocol number 21-0090. All experiments utilized sex- and age-matched mice.

### Generation of Ly49KO mice
The remaining Ly49 receptors in ΔLy49-1 mice were targeted using CRISPR/Cas9 as previously described (*Parikh et al., 2015a*). Briefly, the Ly49 locus was targeted with gRNAs directed against *Klra17* (5′-ACCCATGATGAGTGAGC<u>AGG</u>-3′) and *Klra9* (5′-TGAGACTTCATAAGTCTTCA<u>AGG</u>-3′), with the PAM sequence underlined. For the mRNA microinjections, 20 ng of each guide and 100 ng of Cas9 mRNA were used. Deletions in the Ly49 locus were screened using the primers 5′-GCCCATCT GGCTTCCTTTCT-3′ (*Klra17*-Rv), 5′-CAAGCCCCGATGAGATGGAT-3′ (*Klra9*-Rv), and GGATCAGTCCAT GTCAGGGTT (*Klra17*-Fw) yielding a 409 bp wildtype and a 552 bp mutant band and confirmed using Southern blot analysis (data not shown). To minimize off-target CRISPR/Cas9 effects, candidate founder

mice were backcrossed to C57BL/6 mice for two generations then crossed to derive homozygous Ly49KO mice. Deletions were verified by Sanger sequencing (Azenta Life Sciences) in homozygous Ly49KO mice using the PCR primers *Klra17*-Rv and *Klra9*-Fw for the *Klra17/9* fusion sequence and the primers AACCAAGCCCCAATGAGATC (*Klra7*-Rv) and TGGGTCAGTCCATGTCAGTG (*Klra1*-Fw) for the *Klra7/1* fusion sequence resulting in 552 bp and 409 bp products, respectively.

## Flow cytometry

Fluorescent-labeled antibodies Ly49D (clone 4D11), Ly49EF (CM4), Ly49F (HBF-719), Ly49G2 (eBio4D11), Ly49H (3D10), Ly49I (YLI-90), 2B4 (eBio244F4), CD122 (TM-b1), NKG2AB6 (16a11), NKG2ACE (20D5), CD94 (18d3), NKG2D (CX5), CD27 (LG.7F9), CD11b (M1/70), IFNγ (XMG1.2), CD107a (eBio1D4B), NKp46 (29A1.4), CD3(145-2C11), CD4 (RM4-5), CD8 (53-6.7), TCRB (H597), and CD19 (eBio1D3) were purchased from Thermo Fisher Scientific; Ly49A (YE1/48.10.6), KLRG1 (2F1), and NK1.1 (PK136), were purchased from BioLegend; and Ly49C (4LO311) was purchased from Leinco Technologies. Cells were stained with fixable viability dye eF506 (Thermo Fisher Scientific), continued by staining of cell surface molecules in 2.4G2 hybridoma supernatant to block Fc receptors. For intracellular staining, cells were fixed and stained intracellularly using the BD Cytofix/Cytoperm Fixation/Permeabilization Kit (BD Bioscience) according to the manufacturer's instructions. Samples were acquired using FACSCanto (BD Biosciences) and analyzed using FlowJo software (BD Biosciences). NK cells were defined as singlet Viability-NK1.1+NKp46+CD3-CD19- or Viability-NK1.1+NKp46+CD4-CD8-TCRB-CD19-.

## In vitro stimulation assays

Stimulation of splenic NK cells was performed as previously described (*Parikh et al., 2020*; *Piersma et al., 2019*). Briefly, 1–4 µg/ml anti-NK1.1 (clone PK136, Leinco Technologies) in PBS was coated in 24-well plates for 90 min at 37°C. Plates were washed with PBS and $5 \times 10^6$ splenocytes were added per well. In parallel, splenocytes were stimulated with 200 ng/ml phorbol 12-myristate 13-acetate (PMA; Sigma-Aldrich) and 400 ng/ml ionomycin (Sigma-Aldrich). After 30 min incubation at 37°C, Monensin (Thermo Fisher Scientific) and fluorescently labeled anti-CD107a antibody were added, cultures were incubated for an additional 7 hours at 37°C and subsequently analyzed by flow cytometry.

## In vivo killing assays

In vivo killing assays were performed as previously described (*Parikh et al., 2015b*). Briefly, target splenocytes were isolated from C57BL/6, MHC-I-deficient (TKO), H-2K$^b$-deficient, H-2D$^b$-deficient, KODO and D8-KODO mice. Indicated target splenocytes were differentially labeled with CTV and/or CTFR (Thermo Fisher Scientific). Target splenocytes were additionally labeled with CFSE to identify transferred target splenocytes from host cells. Target cells were mixed at equal ratios for each target and $2 \times 10^6$ splenocytes per target were injected i.v. into naïve hosts. Where indicated, NK cells were depleted with 100 µg anti-NK1.1 (Leinco Technologies) 2 days before target cell injection. Two days after challenge, splenocytes were harvested and analyzed by flow cytometry. Target cell rejection was calculated using the formula $[(1-(\text{Ratio(KO target/wildtype target)}_{sample}/\text{Ratio(KO target/wildtype target)}_{control}))\times100]$.

## Statistics

All experiments were performed at least twice, and representative examples are shown. Cumulative data for two independent experiments is shown for in vivo killing assays. Statistical analysis was performed with PRISM (GraphPad Software) using one-way and two-way ANOVA with corrections for multiple testing. Error bars in figures represent the SEM. Statistical significance was indicated as follows: **** $p<0.0001$; *** $p<0.001$; ** $p<0.01$; * $p<0.05$; ns, not significant.

## Acknowledgements

We thank J Michael White (Transgenic, Knockout, and Micro-Injection Core at Washington University) for CRISPR-Cas9 injections. This work was supported by the National Institutes of Health grants R01-AI129545 to WMY

# Additional information

## Funding

| Funder | Grant reference number | Author |
|---|---|---|
| National Institutes of Health | R01-AI129545 | Wayne M Yokoyama |

The funders had no role in study design, data collection and interpretation, or the decision to submit the work for publication.

## Author contributions

Sytse J Piersma, Conceptualization, Data curation, Formal analysis, Supervision, Investigation, Visualization, Writing – original draft, Writing – review and editing; Shasha Li, Michael D Bern, Data curation, Formal analysis, Investigation, Writing – review and editing; Pamela Wong, Data curation, Formal analysis, Investigation, Writing – original draft; Jennifer Poursine-Laurent, Diana L Beckman, Investigation, Methodology, Writing – review and editing; Liping Yang, Investigation, Writing – review and editing; Bijal A Parikh, Conceptualization, Investigation, Methodology, Writing – review and editing; Wayne M Yokoyama, Conceptualization, Supervision, Funding acquisition, Writing – original draft

## Author ORCIDs

Sytse J Piersma ⬤ https://orcid.org/0000-0002-5379-3556
Shasha Li ⬤ https://orcid.org/0000-0001-5924-6396
Wayne M Yokoyama ⬤ https://orcid.org/0000-0002-0566-7264

## Ethics

This study was performed in strict accordance with the recommendations in the Guide for the Care and Use of Laboratory Animals of the National Institutes of Health. All of the animals were handled according to the approved institutional animal care and use committee (IACUC) protocol (#21-0090). The protocol was approved by the Animal Studies Committee of Washington University.

Reviewer #1 (Public review): https://doi.org/10.7554/eLife.100218.3.sa1
Reviewer #2 (Public review): https://doi.org/10.7554/eLife.100218.3.sa2
Reviewer #3 (Public review): https://doi.org/10.7554/eLife.100218.3.sa3
Author response https://doi.org/10.7554/eLife.100218.3.sa4

# Additional files

## Supplementary files
MDAR checklist

## Data availability

All data generated and analyzed in this study are included in the manuscript and supporting files; source data files have been provided for the figures.

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

# Appendix 1

**Appendix 1—key resources table**

| Reagent type (species) or resource | Designation | Source or reference | Identifiers | Additional information |
|---|---|---|---|---|
| Strain, strain background (*Mus musculus*) | C57BL/6, wildtype | Charles River | Stock# 556 | |
| Strain, strain background (*M. musculus*) | H-2Kb x H-2Db double-deficient | Taconic Farms | Stock# 4215 | |
| Strain, strain background (*M. musculus*) | H-2Dd transgenic mouse | Marguiles | | *Bieberich et al., 1986* |
| Strain, strain background (*M. musculus*) | ΔLy49-1 | Previously generated by authors | | *Parikh et al., 2020* |
| Strain, strain background (*M. musculus*) | Ly49A KI | Previously generated by authors | | *Parikh et al., 2020* |
| Strain, strain background (*M. musculus*) | Ly49 KO | This paper | | *Figure 1A* |
| Sequence-based reagent | *Klra17* gRNA | This paper | gRNA | 5'-ACCCATGATGAGTGAGCAGG-3' |
| Sequence-based reagent | *Klra9* gRNA | This paper | gRNA | 5'-TGAGACTTCATAAGTCTTCAAGG-3' |
| Sequence-based reagent | *Klra17*-Rv | This paper | Sequencing primer | 5'-GCCCATCTGGCTTCCTTTCT-3' |
| Sequence-based reagent | *Klra9*-Rv | This paper | Sequencing primer | 5'-CAAGCCCCGATGAGATGGAT-3' |
| Sequence-based reagent | *Klra17*-Fw | This paper | Sequencing primer | 5'-GGATCAGTCCATGTCAGGGTT-3' |
| Sequence-based reagent | *Klra7*-Rv | This paper | Sequencing primer | 5'-AACCAAGCCCCAATGAGATC-3' |
| Sequence-based reagent | *Klra1*-Fw | This paper | Sequencing primer | 5'-TGGGTCAGTCCATGTCAGTG-3' |
| Antibody | Rat monoclonal anti-human/mouse CD107a eF660 | Thermo Fisher Scientific | Catalog# 50-1071-82 | 1:1000 dilution |
| Antibody | Rat monoclonal anti-mouse/pig CD117 APC | Thermo Fisher Scientific | Catalog# 17-1171-81 | 1:100 dilution |
| Antibody | Rat monoclonal anti-mouse CD11b eF450 | Thermo Fisher Scientific | Catalog# 48-0112-82 | 1:100 dilution |
| Antibody | Rat monoclonal anti-mouse CD122 FITC | Thermo Fisher Scientific | Catalog# 11-1222-82 | 1:50 dilution |
| Antibody | Rat monoclonal anti-mouse CD19 APC-eF780 | Thermo Fisher Scientific | Catalog# 47-0193-82 | 1:100 dilution |
| Antibody | Rat monoclonal anti-mouse CD244.2 (2B4) PE-Cy7 | Thermo Fisher Scientific | Catalog# 25-2441-80 | 1:100 dilution |
| Antibody | Armenian hamster monoclonal anti-human/mouse/rat CD27 APC | Thermo Fisher Scientific | Catalog# 17-0271-82 | 1:100 dilution |
| Antibody | Rat monoclonal anti-mouse CD4 APC-eF780 | Thermo Fisher Scientific | Catalog# 47-0042-82 | 1:100 dilution |
| Antibody | Rat monoclonal anti-mouse CD49b PE | Thermo Fisher Scientific | Catalog# 12-5971-82 | 1:100 dilution |
| Antibody | Rat monoclonal anti-mouse CD8 APC-eF780 | Thermo Fisher Scientific | Catalog# 47-0081-82 | 1:100 dilution |
| Antibody | Rat monoclonal anti-mouse CD94 eF450 | Thermo Fisher Scientific | Catalog# 48-0941-82 | 1:100 dilution |

*Appendix 1 Continued on next page*

*Appendix 1 Continued*

| Reagent type (species) or resource | Designation | Source or reference | Identifiers | Additional information |
|---|---|---|---|---|
| Antibody | Rat monoclonal anti-mouse IFN-gamma eF450 | Thermo Fisher Scientific | Catalog# 48-7311-82 | 1:100 dilution |
| Antibody | Syrian hamster monoclonal anti-mouse/human KLRG1 biotin | BioLegend | Catalog# 138406 | 1:100 dilution |
| Antibody | Syrian hamster monoclonal anti-mouse/human KLRG1 PE-Cy7 | Thermo Fisher Scientific | Catalog# 25-5893-82 | 1:100 dilution |
| Antibody | Rat monoclonal anti-mouse Ly49A FITC | BioLegend | Catalog# 116805 | 1:100 dilution |
| Antibody | Mouse monoclonal anti-mouse Ly49C PE | Leinco | Catalog# L312 | 1:100 dilution |
| Antibody | Rat monoclonal anti-mouse Ly49D APC | Thermo Fisher Scientific | Catalog# 17-5782-82 | 1:100 dilution |
| Antibody | Rat monoclonal anti-mouse Ly49E/F PerCP | Thermo Fisher Scientific | Catalog# 46-5848-82 | 1:100 dilution |
| Antibody | Mouse monoclonal anti-mouse Ly49F PE | BD Biosciences | Catalog# 550987 | 1:100 dilution |
| Antibody | Rat monoclonal anti-mouse Ly49G2 FITC | Thermo Fisher Scientific | Catalog# 11-5781-85 | 1:100 dilution |
| Antibody | Mouse monoclonal anti-mouse Ly49H APC | BioLegend | Catalog# 144710 | 1:100 dilution |
| Antibody | Mouse monoclonal anti-mouse Ly49I Biotin | Thermo Fisher Scientific | Catalog# MA5-28667 | 1:100 dilution |
| Antibody | Mouse monoclonal anti-mouse NK1.1 BV650 | BioLegend | Catalog# 108736 | 1:100 dilution |
| Antibody | Rat monoclonal anti-mouse NKG2A/C/E FITC | Thermo Fisher Scientific | Catalog# 11-5896-85 | 1:100 dilution |
| Antibody | Rat monoclonal anti-mouse NKG2A PerCP | Thermo Fisher Scientific | Catalog# 46-5897-82 | 1:100 dilution |
| Antibody | Rat monoclonal anti-mouse NKG2D PE | Thermo Fisher Scientific | Catalog# 12-5882-82 | 1:100 dilution |
| Antibody | Rat monoclonal anti-mouse NKp46 PE-Cy7 | Thermo Fisher Scientific | Catalog# 25-3351-82 | 1:100 dilution |
| Antibody | Rat monoclonal anti-mouse NKp46 PerCP | ThermoFisher Scientific | Catalog# 46-3351-82 | 1:100 dilution |
| Antibody | Armenian hamster monoclonal anti-mouse TCRB APC-eF780 | Thermo Fisher Scientific | Catalog# 47-5961-82 | 1:100 dilution |
| Antibody | Fc-Block | 2.4g2 hybridoma | In-house produced hybridoma supernatant | Undiluted |
| Antibody | Mouse monoclonal anti-mouse NK1.1 for in vivo use | Leinco | N268 | 100 ug/mouse; 1–4 ug/ml for in vitro stimulation |
| Commercial assay or kit | CellTrace CFSE Cell Proliferation Kit | Thermo Fisher Scientific | Catalog# C34554 | |
| Commercial assay or kit | CellTrace Far Red Cell Proliferation Kit | Thermo Fisher Scientific | Catalog# C34564 | |

*Appendix 1 Continued*

| Reagent type (species) or resource | Designation | Source or reference | Identifiers | Additional information |
|---|---|---|---|---|
| Commercial assay or kit | CellTrace Violet Cell Proliferation Kit | Thermo Fisher Scientific | Catalog# C34557 | |
| Commercial assay or kit | Cytofix/Cytoperm Fixation/ Permeabilization Kit | BD Biosciences | Catalog# 554714 | |
| Chemical compound, drug | Ionomycin calcium salt | MilliporeSigma | Catalog# IO634-5MG | |
| Commercial assay or kit | Monensin | Thermo Fisher Scientific | Catalog# 00-4505-51 | |
| Chemical compound, drug | PMA (phorbol 12-myristate 13-acetate) | MilliporeSigma | Catalog# P1585-1MG | |
| Chemical compound, drug | Sterptavidin-APC | Thermo Fisher Scientific | Catalog# 17-4317-82 | |
| Chemical compound, drug | Sterptavidin-PE | BioLegend | Catalog# 405204 | |
| Chemical compound, drug | eBioscience Fixable Viability Dye eFluor 506 | Thermo Fisher Scientific | Catalog# 65-0866 | |
| Software, algorithm | PRISM 10 | GraphPad | RRID:SCR_002798 | https://www.graphpad.com |
| Software, algorithm | FlowJo 10 | Treestar | RRID:SCR_008520 | https://www.flowjo.com/ |

