## [Editor Report · eLife Assessment]

This study on mouse Ly49 receptors expressed on natural killer (NK) cells shows that Ly49A, in the presence of the corresponding MHC class I allele, can lead to NK cell licensing, thereby providing **valuable** insights into the mechanisms of NK cell modulation by Ly49 receptors. The work may have significant implications for studies of human killer-cell immunoglobulin-like receptors (KIR) expressing and other NK cells. Overall, the study was well-developed with **convincing** evidence.

---

## [Referee Report · Reviewer #1 (Public review)]

Summary:

The article by Piersma et al. aims to reduce the complex process of NK cell licensing to the action of a single inhibitory receptor for MHC class I. This is achieved using a mouse strain lacking all of the Ly49 receptors expressed by NK cells and inserting the Ly49a gene into the Ncr1 locus, leading to expression on all the majority of NK cells.

Strengths:

The mouse model used represents a precise deletion of all NK-expressed genes within the Ly49 cluster. Re-introduction of the Ly49a gene into the Ncr1 locus allows expression by most NK cells. Convincing effects of Ly49a expression on in vitro activation and in vivo killing assay are shown.

Weaknesses:

The choice of Ly49a provides a clear picture of H-2Dd recognition by this Ly49. It would be valuable to perform additional studies investigating Ly49c and Ly49i receptors for H-2b. This is of interest because there are reports indicating that Ly49c may not be a functional receptor in B6 mice due to strong cis interactions. Investigation of the Ly49c and Ly49i receptors in this model would be the basis of future studies that are beyond the scope of the current report.

This work generates an excellent mouse model for the study of NK cell licensing by inhibitory Ly49s that will be useful for the community. It provides a platform whereby the functional activity of a single Ly49 can be assessed.

Comments on revisions: No additional concerns

---

## [Referee Report · Reviewer #2 (Public review)]

Piersma et al. continue to work on deciphering the role and function of Ly49 NK cell receptors. This manuscript shows that a single inhibitory Ly49 receptor is sufficient to license NK cells and eliminate MHC-I-deficient target cells in mice. In short, they refined the mouse model ∆Ly49-1 (Parikh et al., 2020) into the Ly49KO model in which all Ly49 genes are disrupted. Using this model, they confirmed that NK cells from Ly49KO mice cannot be licensed, produce lower levels of IFN-gamma, and cannot reject MHC-I-deficient cells. To study the effect of a single Ly49 receptor in the function of NK cells, the authors backcrossed Ly49KO mice to H-2Dd transgenic KODO (D8-KODO) Ly49A knock-in mice in which a single inhibitory Ly49A receptor that recognizes H-2Dd ligands is expressed. By doing so, they demonstrate that a single inhibitory Ly49 receptor expressed by all NK cells is sufficient for licensing and missing-self killing.

While the results of the study are largely consistent with the conclusions, it is important to address some discrepancies. For instance, in the title of Figure 1, the authors state that NK cells in Ly49KO mice compared to WT mice have a less mature phenotype , which is not consistent with the corresponding text in the Results section (lines 170-171) that states there is no difference in maturation. These differences are not evident in Figure 1, panel D. It is crucial to acknowledge these inconsistencies to ensure a comprehensive understanding of the research findings.

In the legend of Figure 2. the text related to panel C indicates the use of dyes to label the splenocytes, and CFSE, CTV, and CTFR were mentioned. However, only CTV and CTFR are shown on the plots and mentioned in the corresponding text in the Results section. Similarly, in the legend of Figure 4, which is related to panel C, the authors write that splenocytes were differentially labeled with CFSE and CTV as indicated; however, in Figure 4, C and the Results section text, there is no mention of CFSE.

The authors should clarify why they assume that KLRG1 expression is influenced by the expression of inhibitory Ly49 receptors and not by manipulations on chromosome 6, where the genes for both KLRG1 and Ly49 receptors are located. However, a better explanation for the possible influence of other inhibitory NK cell receptors still needs to be included. In the study by Zhang et al. doi: 10.1038/s41467-019-13032-5 the authors showed the synergized regulation of NK cell education by the NKG2A receptor and the specific Ly49 family members. Although in this study, Piersma and colleagues show the control of MHC-I deficient cells by Ly49A+ NKG2A-NK cells in Figure 4., this receptor is not mentioned in the Results or in the Discussion section, so its role in this story needs to be clarified. Therefore, the reader would benefit from more information regarding NKG2A receptor and NKG2A+/- populations in their results.

Comments on revisions: The authors have successfully answered all my questions and edited the manuscript accordingly.

---

## [Referee Report · Reviewer #3 (Public review)]

Summary:

In this study, Piersma et al. successfully generated a mouse model with all Ly49 genes knocked out, resulting in the complete absence of Ly49 receptor expression on the cell surface. The absence of Ly49 expression led to the loss of NK cell education/licensing and consequently, a failure in responsiveness against missing-self target cells. The authors demonstrate the restoration of NK cell licensing by knocking in a single Ly49 gene, Ly49A, in a mouse expressing the H-2Dd ligand for this receptor, which is a novel and important finding.

Strengths:

The authors established a novel mouse model enabling them to have a clean and thorough study on the function of Ly49 on NK cell licensing. Also, by knock in a single Ly49, they were able to investigate the function of a given Ly49 receptor excluding the "contamination" of co-expression any other Ly49 genes. The experiment designing and data interpretation were logically clear and the evidence was solid.

Weaknesses:

The mouse model was somehow genetically similar to a previous study. The experimental work and findings are partially overlapping with the previous work by Zhang et al. (2019), who also performed knockout of the entire Ly49 locus in mice and demonstrated that loss of NK responsiveness was due to the removal of inhibitory, and not activating Ly49 genes.

Potential achievements and discussions: The mouse model developed by the authors holds great potential for advancing NK cell functional studies, particularly regarding the regulation of NK cell functions through receptor-ligand interactions. Moreover, it provides a valuable tool for investigating NK cell education and the development of checkpoint inhibitors. These applications could significantly contribute to the broader research efforts in cancer therapy utilizing NK cells.

Comments on revisions: The authors have successfully addressed all the concerns raised in my previous feedback. They have significantly improved the logical structure, making it clearer and more coherent. Additionally, they have ensured consistency in the use of specific terminology throughout the manuscript. The substantial revisions and re-writing efforts are commendable and have greatly enhanced the overall quality of the manuscript.

---

## [Author Response]

The following is the authors’ response to the original reviews.

**Reviewer #1 (Public review):**
Summary:The article by Piersma et al. aims to reduce the complex process of NK cell licensing to the action of a single inhibitory receptor for MHC class I. This is achieved using a mouse strain lacking all of the Ly49 receptors expressed by NK cells and inserting the Ly49a gene into the Ncr1 locus, leading to expression on the majority of NK cells.Strengths:The mouse model used represents a precise deletion of all NK-expressed genes within the Ly49 cluster. The re-introduction of the Ly49a gene into the Ncr1 locus allows expression by most NK cells. Convincing effects of Ly49a expression on in vitro activation and in vivo killing assay are shown.Weaknesses:The choice of Ly49a provides a clear picture of H-2D^d^ recognition by this Ly49. It would be valuable to perform additional studies investigating Ly49c and Ly49i receptors for H-2b. This is of interest because there are reports indicating that Ly49c may not be a functional receptor in B6 mice due to strong cis interactions.

We agree with the reviewer that it will be important to extend our findings to H-2b haplotypes with individual cognate Ly49 receptors (Ly49C and Ly49I). While these experiments are subject of our ongoing studies, they are beyond the scope of the current manuscript considering the significant time, effort and cost to generate these new Ly49C and Ly49I knockin mice.

This work generates an excellent mouse model for the study of NK cell licensing by inhibitory Ly49s that will be useful for the community. It provides a platform whereby the functional activity of a single Ly49 can be assessed.
**Reviewer #2 (Public review):**
Piersma et al. continue to work on deciphering the role and function of Ly49 NK cell receptors. This manuscript shows that a single inhibitory Ly49 receptor is sufficient to license NK cells and eliminate MHC-I-deficient target cells in mice. In short, they refined the mouse model ∆Ly49-1 (Parikh et al., 2020) into the Ly49KO model in which all Ly49 genes are disrupted. Using this model, they confirmed that NK cells from Ly49KO mice cannot be licensed, produce lower levels of IFN-gamma, and cannot reject MHC-I-deficient cells. To study the effect of a single Ly49 receptor in the function of NK cells, the authors backcrossed Ly49KO mice to H-2D^d^ transgenic KODO (D8-KODO) Ly49A knock-in mice in which a single inhibitory Ly49A receptor that recognizes H-2D^d^ ligands is expressed. By doing so, they demonstrate that a single inhibitory Ly49 receptor expressed by all NK cells is sufficient for licensing and missing-self killing.While the results of the study are largely consistent with the conclusions, it is important to address some discrepancies. For instance, in the title of Figure 1, the authors state that NK cells in Ly49KO mice compared to WT mice have a less mature phenotype , which is not consistent with the corresponding text in the Results section (lines 170-171) that states there is no difference in maturation. These differences are not evident in Figure 1, panel D. It is crucial to acknowledge these inconsistencies to ensure a comprehensive understanding of the research findings.

We thank the reviewer for pointing this out. We have corrected the figure legend title to: “Mice generated to lack all NK-related Ly49 molecules using CRISPR have NK cells that display alterations in select surface molecules.”

In the legend of Figure 2. the text related to panel C indicates the use of dyes to label the splenocytes, and CFSE, CTV, and CTFR were mentioned. However, only CTV and CTFR are shown on the plots and mentioned in the corresponding text in the Results section. Similarly, in the legend of Figure 4, which is related to panel C, the authors write that splenocytes were differentially labeled with CFSE and CTV as indicated; however, in Figure 4, C and the Results section text, there is no mention of CFSE.

We thank the reviewer to point out these inconsistencies. We did label target cells with CFSE to distinguish them from host cells, to clarify we have done the following:

We have removed CFSE from figure legends of Figure 2 and 4.

We included the following on CFSE labeling in the Materials and Methods section: “Target splenocytes were additionally labeled with CFSE to identify transferred target splenocytes from host cells.”

The authors should clarify why they assume that KLRG1 expression is influenced by the expression of inhibitory Ly49 receptors and not by manipulations on chromosome 6, where the genes for both KLRG1 and Ly49 receptors are located.

The effect on KLRG1 expression in phenocopied in the Ly49A KI mice (on a Ly49 KO background). The Ly49A KI allele is encoded by the Ncr1 locus, which is located on chromosome 7 and not by chromosome 6 where KLRG1 is located, thus excluding involvement of cis-regulatory elements encoded by the *Ly49* locus on chromosome 6.

We have clarified this in the discussion section (lines 350-358):

“The *Ly49* gene family as well as *Klrg1* is located within the NKC on chromosome 6 (Yokoyama and Plougastel, 2003) …. expression of only Ly49A, encoded in the *Ncr1* locus located on chromosome 7, in Ly49KO mice on a H-2D^d^ background restored KLRG1 expression”

However, a better explanation for the possible influence of other inhibitory NK cell receptors still needs to be included. In the study by Zhang et al. doi: 10.1038/s41467-019-13032-5 the authors showed the synergized regulation of NK cell education by the NKG2A receptor and the specific Ly49 family members. Although in this study, Piersma and colleagues show the control of MHC-I deficient cells by Ly49A+ NKG2A-NK cells in Figure 4., this receptor is not mentioned in the Results or in the Discussion section, so its role in this story needs to be clarified. Therefore, the reader would benefit from more information regarding NKG2A receptor and NKG2A+/- populations in their results.

We agree with the reviewer that it is important to describe our results in the context of other inhibitory receptors. To clarify the role of NKG2A and potentially other inhibitory receptors we have made the following improvements to our manuscript:

We discuss the role of NKG2A in the discussion section, which now include (lines 259-266):

“While our results did not interrogate licensing by inhibitory receptors outside of the Ly49 receptor family, such as has been reported for NKG2A (Anfossi et al., 2006; Zhang et al., 2019), they do demonstrate that expression of Ly49A without other Ly49 family members can mediate NK cell licensing. Moreover, we found that Ly49 receptors are required and sufficient for missing-self rejection under steady-state conditions. However, these observations do not rule out involvement of other inhibitory receptors under specific inflammatory conditions. For example, NKG2A contributes to rejection of missing-self targets in poly(I:C)-treated mice (Zhang et al., 2019).”

We also added the following to the result section (lines 179-182):

NKG2A has been implicated in NK cell licensing by the non-classical MHC-I molecule Qa1 (Anfossi et al., 2006), to eliminate potential confounding effects by this interaction, effector functions of NKG2A- NK cells were evaluated as described before (Bern et al., 2017).

**Reviewer #3 (Public review):**
Summary:In this study, Piersma et al. successfully generated a mouse model with all Ly49n et al., 2017 genes knocked out, resulting in the complete absence of Ly49 receptor expression on the cell surface. The absence of Ly49 expression led to the loss of NK cell education/licensing and consequently, a failure in responsiveness against missing-self target cells. The experimental work and findings are partially overlapping with the previous work by Zhang et al. (2019), who also performed knockout of the entire Ly49 locus in mice and demonstrated that loss of NK responsiveness was due to the removal of inhibitory, and not activating Ly49 genes. The authors demonstrate the restoration of NK cell licensing by knocking in a single Ly49 gene, Ly49A, in a mouse expressing the H-2D^d^ ligand for this receptor, which is a novel and important finding.Strengths:The authors established a novel mouse model enabling them to have a clean and thorough study on the function of Ly49 on NK cell licensing. Also, by knocking in a single Ly49, they were able to investigate the function of a given Ly49 receptor excluding the "contamination" of co-expression of any other Ly49 genes. Their idea and method were novel though the mouse model was somehow genetically similar to a previous study. The experiment design and data interpretation were logically clear and the evidence was solid.Weaknesses:The paper is very poorly written and confusing. The authors should be more accurate in the usage of terminology, provide more details on experimental procedures, and revise much of the text to improve clarity and coherence. A thorough revision aiming to clarify the paper would be helpful.

We regret that the manuscript was confusing to the reviewer. We have made thorough revisions to the different sections, which we hope will improve the clarity of the manuscript.

We have made changes to all sections of the manuscript, including the title. These revisions include improved clarity on description of NK cell licensing and consistent usage throughout the manuscript, per the reviewer recommendations. We hope that all our improvements help the clarity of the manuscript.

**Recommendations for the authors:**

**Reviewer #1 (Recommendations for the authors):**
I was confused by lines 262-270 in the discussion. The data from Hanke et al. is presented as contradictory to the observation that Ly49s bind more efficiently to H2-Kb than -Db, but they showed that Ly49c/i did not bind Kb-deficient cells, supporting the preferred binding to Kb.

We have clarified this issue and the paragraph now reads: “This is further supported by early studies using Ly49 transfectants binding to Con A blasts showing that Ly49C and Ly49I can bind to H-2D^b^-deficient but not H-2K^b^-deficient cells (Hanke et al., 1999), despite the caveat of testing binding to cells overexpressing Ly49s in these studies.”

**Reviewer #2 (Recommendations for the authors):**
The authors' conclusion that one type of inhibitory Ly49 receptor expressed on NK cells is sufficient for successful licensing and rejection of missing self-cells is a significant step forward. However, it would be beneficial to complement this with additional data. For instance, exploring the role of a single inhibitory Ly49 receptor responsible for licensing in a mouse model with a different haplotype (e.g. Ly49C or Ly49I on H-2b MHC I haplotype in C57BL/6J mice) could provide valuable insights and open new avenues for research in the field.

We agree with the reviewer that it will be important to extend our findings to additional MHC-I haplotypes with single cognate Ly49 receptors. While these experiments are subject of our ongoing studies, they are beyond the scope of the current manuscript considering the significant effort, time, and cost to generate these new Ly49C and Ly49I knockin mice.

**Reviewer #3 (Recommendations for the authors):**
Specific issues that should be addressed are as follows:(1) The title of the paper: "Expression of a single inhibitory Ly49 receptor is sufficient to license NK cells for effector functions" is ambiguous. When I first read the title, I thought the authors meant that only a single Ly49 molecule on the NK cell surface was necessary to induce licensing. It might be better to replace "single inhibitory receptor" with "single member of Ly49 receptor family".

We have changed the title to: “Expression of a single inhibitory member of the Ly49 receptor family is sufficient to license NK cells for effector functions”

(2) In the abstract, introduction, and results, the authors distinguish "licensing" and "rejection of missing-self targets" as two distinct phenomena. An example includes Abstract, lines 51-53: "Herein, we showed mice lacking expression of all Ly49s were unable to reject missing-self target cells in vivo, were defective in NK cell licensing, and displayed lower KLRG1 on the surface of NK cells". Similarly, the title of the second subsection of the Results states: "Ly49-deficient NK cells are defective in licensing and rejection of cognate MHC-I deficient target cells" (line 176). In these instances, it seems that by "licensing", they mean only response to plate-bound anti-NK1.1 stimulation and not a response to missing-self targets. Alternatively, in the first paragraph of the Discussion, it sounds as if licensing includes both anti-NK1.1 and missing-self responses (lines 258-260): "...NK cells were fully licensed in terms of their functional phenotype, including the capacity to be activated by an activation receptor in vitro and efficient rejection of MHC-I deficient target cells in vivo". Please define the terms and use the terms consistently throughout the paper.

We were the first to describe the term licensing and have defined this as acquisition of NK cell functional competence by self-MHC molecules (Kim et al., 2005), which is characterized by increased NK cell effector functions to activating signals. Thus, licensed NK cells are prevented from attacking normal MHC-I^+^ cells by the same self-MHC-I-specific receptor that conferred licensing, while unlicensed NK cells without appropriate Ly49 receptors are functionally incompetent.

To clarify we made changes throughout the manuscript including the following:

Lines 91-101:

“In addition to effector function in missing-self, Ly49 receptors that recognize their cognate MHC-I ligands are involved in licensing or education of NK cells to acquire functional competence. NK cell licensing is characterized by potent effector functions including IFNγ production and degranulation in response to activation receptor stimulation (Elliott et al., 2010; Kim et al., 2005). Like missing-self recognition, inhibitory Ly49s require SHP-1 for NK cell licensing which interacts with the ITIM-motif encoded in the cytosolic tail of inhibitory Ly49s (Bern et al., 2017; Kim et al., 2005; Viant et al., 2014). Moreover, lower expression of SHP-1, particularly within the immunological synapse, is associated with licensed NK cells (Schmied et al., 2023; Wu et al., 2021). Thus, inhibitory Ly49s have a second function that licenses NK cells to self-MHC-I thereby generating functionally competent NK cells but it has not been possible to exclude contributions from other co-expressed Ly49s.”

Lines 268-271 (previously 258-260):

“Yet the NK cells were fully licensed in terms of IFNγ production and degranulation in vitro and efficiently rejected MHC-I deficient target cells in vivo. Thus, a single Ly49 receptor is capable to confer the licensed phenotype and missing-self rejection in vitro and in vivo.”

Lines 309-312:

“In conclusion, these data show that expression of a single inhibitory Ly49 receptor is necessary and sufficient to license NK cells and mediate missing self-rejection under steady state conditions in vivo.”

(3) Introduction, lines 76-79. Please provide the C57BL/6 MHC-I genotype. It is difficult to follow the text here without this information. In general, please provide information to help the reader who may not be working in this precise field.

We thank the reviewer for pointing this out. We have included this and the lines now read: “For example, in the C57BL/6 background, Ly49C and Ly49I can recognize H-2^b^ MHC-I molecules that include H-2K^b^ and H-2D^b^, while Ly49A and Ly49G cannot recognize H-2^b^ molecules and instead they recognize H-2^d^ alleles.”

(4) Introduction, lines 85-97. Please use commas: "...the MHC-I specificities of other Ly49s have been primarily studied with MHC tetramers containing human b2m, which is not recognized by Ly49A, on cells overexpressing Ly49s" in order to clarify the sentence.

Commas have been added as suggested by the reviewer.

(5) Introduction, lines 91-101. The whole paragraph starting with the following sentence does not make sense and should be re-written. "In addition to effector function in missing-self, when inhibitory Ly49 receptors recognize their cognate MHC-I ligands in vivo, they license or educate NK cells for potent effector functions including IFNγ production and degranulation in response to activation receptor stimulation".

We regret that this paragraph was not clear to the reviewer. We have changed this paragraph to:

“In addition to effector function in missing-self, Ly49 receptors that recognize their cognate MHC-I ligands are involved in licensing or education of NK cells to acquire functional competence. NK cell licensing is characterized by potent effector functions including IFNγ production and degranulation in response to activation receptor stimulation (Elliott et al., 2010; Kim et al., 2005). Like missing-self recognition, inhibitory Ly49s require SHP-1 for NK cell licensing which interacts with the ITIM-motif encoded in the cytosolic tail of inhibitory Ly49s (Bern et al., 2017; Kim et al., 2005; Viant et al., 2014). Moreover, lower expression of SHP-1, particularly within the immunological synapse, is associated with licensed NK cells (Schmied et al., 2023; Wu et al., 2021). Thus, inhibitory Ly49s have a second function that licenses NK cells to self-MHC-I thereby generating functionally competent NK cells but it has not been possible to exclude contributions from other co-expressed Ly49s.”

(6) Results, line 181. Please edit: "...MHC-I-deficient H-2K^b^ x H-2D^b^ deficient (KODO) mice".

This sentence now reads “... NK cells from H-2K^b^ and H-2D^b^ double deficient (KODO) mice”

(7) Results, line 192. Please re-word the following phrase: "missing-self is dominated by H-2K^b^ in the C57BL/6 background", as it is unclear. Do you mean that H-2K^b^ is protected from lysis as opposed to H-2D^b^?

We thank the reviewer for pointing this out, line 192 now reads: “..missing-self recognition in the C57BL/6 background depends on the absence of H-2K^b^ rather than H-2D^b^.”

(8) Please briefly describe the Ncr1-Ly49A knockin procedure so that the reader understands the link between NKp46 and Ly49A expression without going to the earlier paper. Also, it needs to be mentioned that Ncr1 is the gene encoding NKp46.

Lines 201-205 now read: “To investigate the potential of a single inhibitory Ly49 receptor on mediating NK cell licensing and missing-self rejection, the Ly49KO mice were backcrossed to H-2D^d^ transgenic KODO (D8-KODO) Ly49A KI mice that express *Klra1* cDNA encoding the inhibitory Ly49A receptor in the *Ncr1* locus encoding NKp46 and its cognate ligand H-2D^d^ but not any other classical MHC-I molecules (Parikh et al., 2020).

In the materials and Methods section, the following has been added (lines 324-326):

“In Ly49A KI mice the stop codon of Ncr1 encoding NKp46 is replaced with a P2A peptide-cleavage site upstream of the Ly49A cDNA, while maintaining the 3’ untranslated region.”

(9) Figure 4C, legend. There is no CFSE staining in this experiment. Please correct.

We did label target cells with CFSE to distinguish them from host cells, to clarify we have done the following:

We have removed CFSE from figure legends of Figure 2 and 4.

We included the following on CFSE labeling in the Materials and Methods section (lines 377-379): “Target splenocytes were additionally labeled with CFSE to identify transferred target splenocytes from host cells.”

(10) Discussion, lines 262-270. This paragraph sounds as if data by Hanke et al. does not agree with the data presented in the paper. On the contrary, Hanke et al. demonstrate that Ly49C and Ly49I detectably bind to H-2K^b^, but poorly to H-2D^b^, supporting observations shown in Figure 2C.

We have clarified this issue and the paragraph now reads: “This is further supported by early studies using Ly49 transfectants binding to Con A blasts showing that Ly49C and Ly49I can bind to H-2D^b^-deficient but not H-2K^b^-deficient cells (Hanke et al., 1999), despite the caveat of testing binding to cells overexpressing Ly49s in these studies.”